# The Multilevel Knowledge Economy Pyramid Model as a Flexible Solution to Address the Impact of Adverse Events in the Economy

**Octavian Șerban**

Department of Regional Development, Bucharest University of Economic Studies, 010374 Bucharest, Romania; octavian.serban@eam.ase.ro

**Abstract:** The complexity of the current crisis is a big challenge for sustainable development, in the context of several overlapping shocks generated by the Coronavirus disease, geostrategic conflict, energy breaches, and food security threats. An appropriate answer to address these issues is to create a new approach to sustainable growth based on the knowledge economy. At this point, related studies have referring to the knowledge triangle, the triple (quintuple) helix, smart specialization, and the knowledge-based economy. In this research, compared with other studies in the field, knowledge structures created on the foundation of the knowledge economy were designed to work together to generate synergy in a knowledge environment where the stakeholders are universities, technology providers, governments, local communities, and entrepreneurs. In this mechanism, innovation, creativity, and entrepreneurship are the premises for increasing productivity and competitiveness, with a positive impact on smart growth and the standard of living.

**Keywords:** knowledge economy; productivity; competitiveness; innovation; cris

## 1. Introduction

The human being is the root of embedded knowledge in the production flow and technology is merely an enabler, accelerator, and amplifier of knowledge diffusion in all the processes, both down- and upstream. From this perspective, it can be said that intellectual capital is the main contributor to increasing productivity. It is important to underline the complexity of knowledge in highly performant processes: the mixture of humans and machines, innovation and accuracy, creativity and quality. The importance of original knowledge ownership, related to humans, can be overcome by the power of knowledge generated in the production flow through the contribution of several other factors such as the following: technology, digital advancement, management capability, and others [1].

Simple knowledge related to low skills and education, routine work, and rigid flows makes a small contribution to productivity. Amplified process knowledge is the main contributor to progress, and this embedded knowledge is, in fact, the total factor productivity (TFP) [2]. This is quite difficult to measure in a simple way with a "magic formula", but important steps have been taken in the last decade. From this perspective, it is commonly agreed that TFP is a residual in the production function.

During the pandemic crisis, as restrictions were in place in different shapes, amplitudes, and intensities, global concern persists regarding productivity as the cornerstone of sustainable growth in income and keeping the standard of living to the same level, at least.

From this perspective, besides some improvements in regulations, organizational flows, and the digital environment, this moment is quite opportune for rethinking the activity around knowledge as an asset and a resource at the same time. Knowledge management is a methodology which can help in arranging production flows around knowledge, and in developing a knowledge process, knowledge communities, and value-added.

Rethinking the working process around knowledge has some urgency in this new challenging environment of remote work activities. Similar to with the teleworking regime, knowledge methodology was very well developed before the current crisis, but the urgency of the moment could accelerate the implementation of knowledge management in every organization. Based on an efficient use of knowledge, according to the methodology, it is possible to create knowledge workers and knowledge organizations, and ultimately to build up what is called a knowledge economy [3].

Smart and sustainable growth is strongly considering innovation, intellectual capital enhancement, and TFP. At the same time, in this context, it is necessary to consider diverse stakeholders, knowledge environments, and supporting organizations.

In the current geostrategic context of the pandemic crisis, energy disruptions, and food scarcity, the role of governments in economy is increasing, and economic and social development is directly linked to the collaboration of the key actors. The role of governments is to mitigate the influence of external factors which affect the economic and social environment, such as health in terms of the COVID-19 pandemic crisis, energy in terms of political decisions regarding Russia, conflict in terms of wartime conditions, food supply in terms to crops, and so on.

Several years ago, productivity was considered from an engineering, technical, or economic perspective, and was related to the potential of equipment, the effectiveness of processes, and the efficiency of a business. Now, productivity and competitiveness issues are moving into the fields of innovation, creativity, and increasing the value of knowledge.

The evolution of global context, during the end of 2019 to mid-2022, moved the complexity of the context from competition in the market to the fight against COVID-19, changes in the energy suppliers' geopolitical strategies, and facing the threat of war.

In this context, the business environment was dramatically affected, and collaborative work is needed at the economic and social levels in order to keep the economies alive, continue the progress of productivity, and reshape the conditions for competitiveness.

It is quite difficult to identify the effects of each crisis driver: disease, energy disruption, war, and food security. This research started by analyzing the effects of the COVID-19 crisis on the economy, but now, other shocks have developed: energy disruptions, geopolitical conflict, and food security crises.

This enlarged perspective of crisis will change the response to sanitary threats, strategic energy supply, geopolitical cooperation, and alternative food resources.

## 2. Literature Review

### 2.1. Definitions of the Key Concepts of the MKEP Model

The knowledge economy concept is the process of creating knowledge in order to deliver real benefits for the population in terms of the best performance in increasing productivity and competitiveness [4].

The triple helix concept represents the mechanism of interconnecting the university as a leader of the knowledge transformation process with governments and the business environment in one ecosystem that generates new knowledge [5].

The knowledge triangle concept was designed to explore the optimal solutions for better alignment of research, education, and innovation [6].

Smart growth is a concept of decoupling economic development through the extensive use of resources, especially non-renewable ones, and addresses strategies for preserving the natural environment and protecting health to make the role of communities stronger from the economic, social, and cultural perspective. "Smart growth" covers a range of development and conservation strategies that help protect our health and natural environment, and make our communities more attractive, economically stronger, and more socially diverse [7].

Smart specialization is the process of spreading knowledge horizontally with the purpose of increase knowledge at the regional level within universities, industry, and administration, for better integration of knowledge [8].

## 2.2. Evolutionary Process of the Collaborative Knowledge Environment

In previous research [9], the importance of tacit knowledge was already emphasized in the production process, and awareness of the transformation of knowledge from tacit to explicit and to tacit again, in a loop, has been very well represented in the SECI model [10].

The multilevel knowledge economy pyramid represents an upgrading knowledge environment where increasing productivity is a key driver for enhancing the quality of life. At this point, it is necessary to try to evaluate whether the pandemic shock has affected productivity, as well as the trends during the crisis and the perspectives for the next few years.

## 2.3. Correlation between Productivity and Crisis

According to the US Bureau of Labor Statistics [11], in the last 15 years, productivity decreased compared with the previous 10 years, and the average growth level was 1.3%. This low performance is partially due to the financial crisis of 2008–2010. During the COVID-19 shock, the same source reported that productivity increased by 5.4% at the beginning of 2021. Even so, the adverse effects spread throughout the global economy. For example, the GDP of advanced economies was more affected than that in emergent economies, given a drop of more than 2% between these two types of economies. According to the same source, as one can notice in Figure 1, in the last two decades, labour productivity has fluctuated but the trend has constantly decreased. During the financial crisis of 2008–2010 and the pandemic crisis, the revival was significant. Both times, governments were invited to inject huge financial support to overcome the adverse effects on the global economy, especially to protect jobs.

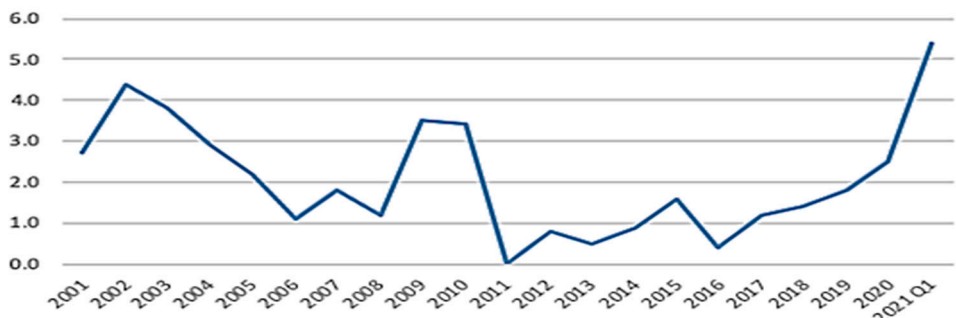

**Figure 1.** Annual labour productivity growth, 2001–2021 Q1. Source: US Bureau of Labor Statistics.

The inputs for improving the affected processes were generated by the context of the pandemic itself. It is considered that the deployment of digitalization on a large scale, enlargement of the teleworking regime, the massive allocation of government resources, and many other actions overcame the negative effects of the pandemic crisis and reversed the performance trend.

## 2.4. Finding Solutions to Cope with Adverse Events in the Economy

One way to overcome hard times during crises is to put digitalization in the first place, with great impact on communication among people working from home. This was an efficient formula which succeeded in overcoming the inconvenience of lower mobility due to the restrictions imposed. During the evolution of the pandemic, in terms of waves, vaccination, and medication, the restriction regimes changed. Under these conditions, the necessity of enlarging the contribution of digitalization [12] to the development of socio-economic life was a great opportunity and needs to be permanently improved in the years to come.

Adopting of digitalization, spreading innovation, and moving into a new age of technology based on artificial intelligence—in other words, moving closer or beyond the knowledge frontier—is the most reliable development strategy for knowledge-based economies [13] By accepting these new conditions of competition in the virtual space, with

the latest technology and remote labour availability, all these contribute to the ability to use knowledge as a resource and as an endogenous factor in the production function [14].

Beside macroeconomic stability, free trade is a premise for good results in terms of economic growth. It is already a fact that countries with open economies have the best results in the productivity field. Free trade in complex economic environments contributes to the diversification of production, the best transfer of the latest technologies, free movement of labor, and easy access to the international market and resources. This process is a cornerstone for complex goods, excellent value added in production, technological alignment, knowledge-intensive processes, and high-level skills. All these are basic components of increased productivity, sustainable development, competitiveness, and a high standard of living [15].

It has been well documented that once an economy becomes open and foreign direct investment enters the country, the overall productivity rate improves as a result of new technology that works on the spot, more organized production flows, highly trained people, and the possibility of producing for the export market.

By organizing all these drivers [16], it can be observed that higher productivity is determined by several factors:

- Internal factors—improved management skills, adoption of new methodologies for quality improvement, acquisition of new technologies, investment in human capital, process knowledge as a resource, innovating, encouraging creativity, increased entrepreneurial attitudes;
- External factors—fostering collaborative organizational behavior, improving legislation, increased administrative capacity, building integrated research—education—business strategies;
- Macroeconomic factors—free trade, an open economy, foreign direct investment, real integration in the global value chain.

During the shocks, some of these drivers were dramatically affected, especially those coming from the external environment or those at the macroeconomic level. Here, studies identified the value of the government as a strategic partner [17] to take rapid action to mitigate all the threats and to substantially change the business environment in a way that supported the activities of the economic and social actors.

At this point, the pandemic crisis is an example of a dramatic change in the business environment and the prompt response of the governments, which also includes the role of the European Commission, which have generated the possibility of a rapid recovery. Huge public budget allocations for supporting the activity of firms, using subsidies to maintain jobs, modifying state aid rules, relaxing the fiscal burden, and many other levers are just some of direct governmental interventions in the most affected business ecosystems.

This prompt response was planned to work in the short term to give enough time to regain the economic pace existing before any crisis. The critical point of an actual decline in development is the overlapping effect of several crises, such as pandemics, energy supply, war, and food security. This inflection point in global development could also generate a financial crisis, unemployment, disruptions in the global supply chain, etc. Some of the premises for these extended crises have already shown up; for example, the disruption of the global supply chain had affected already the automotive industry and other manufacturing industries [18].

### 2.5. The Impact of Knowledge on Productivity

At the individual level, the growth rate of hourly productivity has decreased, especially in developed economies. This reflects an obvious pressure on individual incomes with an impact on the standard of living. Certainly, the productivity of the other production factor, capital, could be considered, which is very important, but its contribution is also declining. Overall, the growth rate of productivity has decreased. Solow revealed that technological advancement is the accelerator of the productivity growth rate [19]. Moreover, Romer published the endogenous growth theory, in which knowledge is a resource in the

production flow [20]. Despite these findings, modifying the trend of the productivity growth rate is a great challenge of nowadays, with efforts being made to rethink and reshape business models.

In this context of strengthening the knowledge–productivity correlation, there are some measures that can be taken to increase the skills and the level of education, improve technology, and increase the intensity of the processes. These drivers are determinants of increased competitiveness. However, besides these, it is necessary to reconsider the power of knowledge. In order to make a better assessment, it is necessary to simplify all processes to processes of knowledge and to try to find the optimal way to generate knowledge value-added. According to this approach, knowledge is used in the process of innovation, creativity, and entrepreneurship, with the aim of increasing productivity and competitiveness, ultimately improving the standard of living [21].

## 3. Research Methodology

This work is exploratory research with the objective of rearranging the knowledge structures functioning already in the economy, considering the foundation of the knowledge economy. In this new approach, the visual representation of the multilevel knowledge economy pyramid (MKEP )will validate a new model of development using knowledge, innovation, and creativity to increase productivity and competitiveness, thus increasing the capacity to cope better with adverse events in the economy.

In this context, it is important to build up a more collaborative, flexible, and sustainable environment based on knowledge and the synergy created among the stakeholders, such as universities, companies, technology providers, government, local communities, and others.

Based on the four pillars of the knowledge economy, the new knowledge environment consists of very strong knowledge structures, such as the knowledge triangle, the triple helix, and smart specialization. Some authors have made consistent contributions to the "quintuple helix" cooperation framework [22], but it can be observed that our structure includes all five elements: universities (matching Pillar I of the knowledge economy's foundations), the government (matching Pillar IV of the knowledge economy's foundations), industry (matching the level of productivity where education meets the business environment at Level 2), civil society (matching Pillar II of the knowledge economy's foundation), and the environment (matching the intersection of the triple helix and smart growth). It was redundant to mention the quintuple helix in our model, as the layout of the key elements allowed us to make the connection with any elements we wanted or needed [23]. The value of the MKEP model is that it allows a compressed and easily understood visual representation of all the actors involved and the connections between them. On the other hand, this format was chosen because the entire pyramid involves three different perspectives of the development model: three levels built upon a knowledge foundation, three structures, and three angles within each structure. This approach could help to simplify all the elements of progress in research in the field, resulting in a robust and deeply functional model. It can easily be observed that these three structures have a common intersection, the vertical axis of knowledge, which forms the connection between education, represented by the university, and welfare, represented by the standard of living, as can be seen in Figure 2.

The prosperity of the population is the peak of the knowledge system created here as a sustainable alternative model of development. To achieve a high level of quality of life, it is necessary to establish intermediary levels upon the knowledge economy foundation, and these levels are as follows:

- The first level: knowledge foundation. The four pillars of the knowledge economy have education, represented by universities, at the center of the foundation; at this level, knowledge is enriched to the status of input to the next levels;
- The second level: productivity. This is the intersection of knowledge coming from academia and the business environment. When knowledge is included in a value-added process, the transfer of knowledge from the university to the business environ-

ment generates a boost in productivity. At the next level, competitiveness is increasing, not in the traditional "winner takes it all" way but in a collaborative way called "collaborative complementarity"; i.e., creating a knowledge ecosystem where the business environment interacts with the education field with input from research organizations and technology providers, having the goal of enriching the knowledge of the process, and the result is increased productivity;

- The third level: competitiveness. This is the intersection of the knowledge axis improved by productivity (previous level) with smart growth. Once the knowledge has been enriched at the first level, the second stage creates the synergy among knowledge stakeholders (universities, companies, and research and technology providers on the one hand, and governments and local communities on the other hand). This synergy is able to move the process to the next step, closer to what is called the "knowledge frontier", and the cycle restarts with value-added knowledge following the same process at a higher level of performance.
- The fourth level: welfare. This is the result of the evolving process of the previous levels. Analysing economic processes and coming up with complex solutions does not make any sense if the population's interests are not addressed. Referring only to efficiency, effectiveness, process engineering, higher productivity, and solid competitiveness is incomplete if the development models do not consider how the product is shared and how this contributes to the welfare of individuals.

The foundation of this model is the knowledge economy with its four pillars, according to well-known international research in the field [25]: education and training, the ITC infrastructure, economic incentives and institutional support, and innovation systems.

Before the health crisis, in the productivity figures from Eurostat, a constant decline in performance can be observed, as labour productivity at the global level declined after 2007 (the last year before the previous financial crisis) which reached 2.8%, in contrast to 1.4% in 2016. Up until the pandemic crisis, performance increased slowly, but remained below a value of 2%.

By analysing the international context in terms of weak productivity performance, it is possible to identify some critical issues:

- Reallocating resources for avoiding unemployment, reducing the working hours, and providing subsidies for most affected sectors have led to a lack of resources for investment. The sense of urgency during the pandemic drove entrepreneurs, companies, and governments to address critical issues, especially those related to the direct effects of the crisis.
- Reducing mobility and effectively suspending transportation activities created a downsizing of the trade sector, even though online commerce had accelerated its path.
- The possibility of rearranging the workforce around the most efficient sectors was hampered by the restrictions imposed in the area of mobility on the one hand, and the growth of unemployment in a very short time on the other hand; recalibrating skills in such a short time is not an easy job for neither employers nor employees.
- In the long-term, the disruption of educational programs will create some breaches regarding the accumulation of the knowledge necessary to go further in the educational process.
- A breach in the global supply chain has affected important industries. When the outcomes of production flows were diminished, the demand side was not covered and returns suffered.
- Allocating resources to cover the side-effects of the pandemic crisis put more pressure on governmental debts by increasing structural deficits.
- Bottlenecks were created because of limitations in the digital infrastructure and the possibility of regaining intensity takes time.

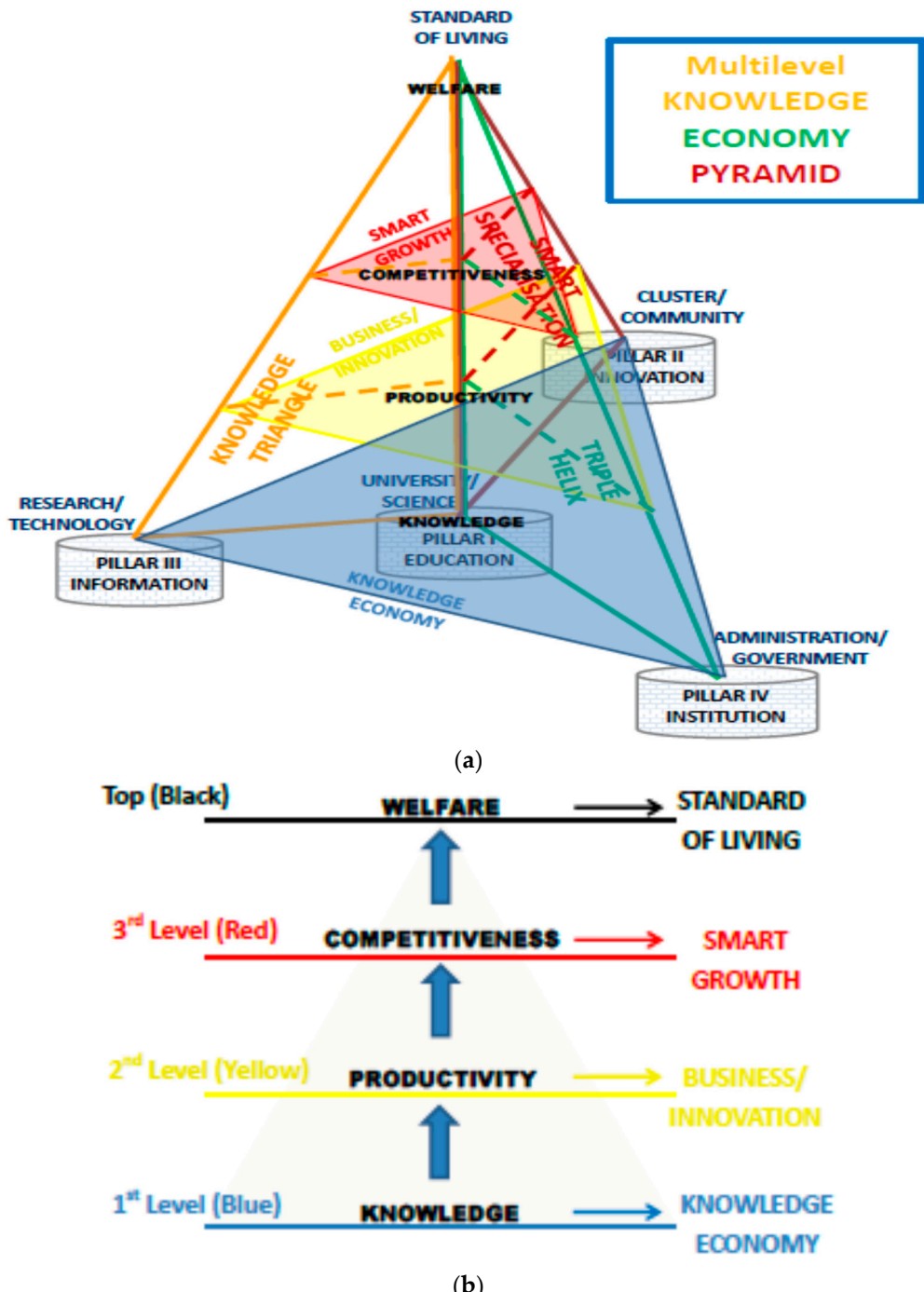

**Figure 2.** Visual representation of the multilevel knowledge economy pyramid: (**a**) 3D representation of the MKEP model; (**b**) MKEP model in section. Source: Author's own work [24].

Governments were deeply involved in this battle to regain economic and social momentum, and some actions were performed to mitigate the obstacles facing increased productivity, such as:

- Shaping policies to accelerate investment, especially in the research, education, and innovation fields on the one hand, and transportation, digital infrastructure, and reindustrialization on the other hand;
- Creating premises for better adaptation to potentially new crises, especially in the energy sector, rearranging the energy suppliers' maps and circuits, investing in renew-

able energy, mitigating climate change, and focusing on knowledge-intensive value creation processes, instead of labour-intensive processes;

- Reshaping the development model around the knowledge-based economy concept, better understanding the dynamics of the knowledge society, and implementing collaborative models where complexity arises and the cooperation between different ecosystems is needed;
- Facilitating the creation and consolidation of flexible working environments; finding alternative ways of conducting business, particularly the more connected with the digital arena; and stimulating the SME sector;
- Recalibrating and readjusting the education system through a strong connection with the real economy and concrete necessities of the business environment.

Searching for solutions to solve productivity issues occurring at a certain moment is an adequate response, but for a proactive attitude, the productivity landscape has to be enlarged to anticipate the inputs from the education, research, governance, capacity building, intellectual capital, investment strategy, and technological fields. In times of shock, this widened framework is even more necessary. Basically speaking, disruptive factors such as the pandemic, rising energy costs, food security, and the geopolitical crisis are very destructive for growth but create some opportunities to rethink productivity in terms of a new framework.

It is necessary to mention here that at beginning of 2022, the most innovative organizations performed the best during the crisis, and Apple became the first U.S. company to reach a market capitalization of USD3 trillion. The same trend was observed for Tesla, SpaceX, Alphabet, Amazon, and many other knowledge organizations. Considering the Schumpeterian "creative destruction" theory and the "cleansing effect", it can be observed that resources were reallocated to the Big Tech businesses. The global community is ready to use innovation as a measurement standard to find the most appropriate business solutions to satisfy their needs. The MKEP model is revealing, in a very understandable way, the supremacy of innovation, research, creativity, and entrepreneurship for capitalizing on the resources of those companies which promote innovation as the main activity.

Solving productivity issues and the declining trend of the "business as usual" mode is a concern which goes beyond the factory or production line. Today, governments and entrepreneurs are partners in these efforts. Strengthening institutional capacity, reshaping legislation, increasing public investment allocations, and creating cooperation platforms are just some of the key drivers of increased competitiveness in such a complex international environment.

Enhancing innovation is a solution for long-term development to reverse the trend of productivity. Innovation has become a differentiator in competitiveness. Investing in innovation is a key input to sustainable growth. Research and development activities are the foundation for boosting innovation. The diffusion of technology and increased capacity to absorb new technologies contribute to increased innovation and thus to improved productivity.

From this perspective, the productivity approach in emergent countries is still different from that of well-developed countries. At this point, it could be said that emergent countries have to consider the accumulation of capital as a source of productivity growth, while developed countries are more reliant on qualitative endogenous growth factors, such as knowledge, innovation, and research. Certainly, both approaches are not exclusive to each other; the best way is to create a mix for increasing productivity.

In other words, the best approach is to combine quantity and quality inputs in the production flow. Those countries situated far from the technology frontier have to rely more on quantitative inputs and first develop a strong infrastructure, then consider adopting a qualitative-based input strategy when the gap has decreased and the capacity for absorption of the new technologies is in place.

The catching-up factor in this rationale is the capacity to compete in the global arena and to adopt new technologies on the market. Free trade and removing the obstacles facing

international exchange have a positive effect on the adoption of new technologies, then on increased innovation, ultimately improving productivity.

During the last few decades, in terms of changes in productivity trends, there was a transformation in the sectors of activity, as it is presented in Figure 3.

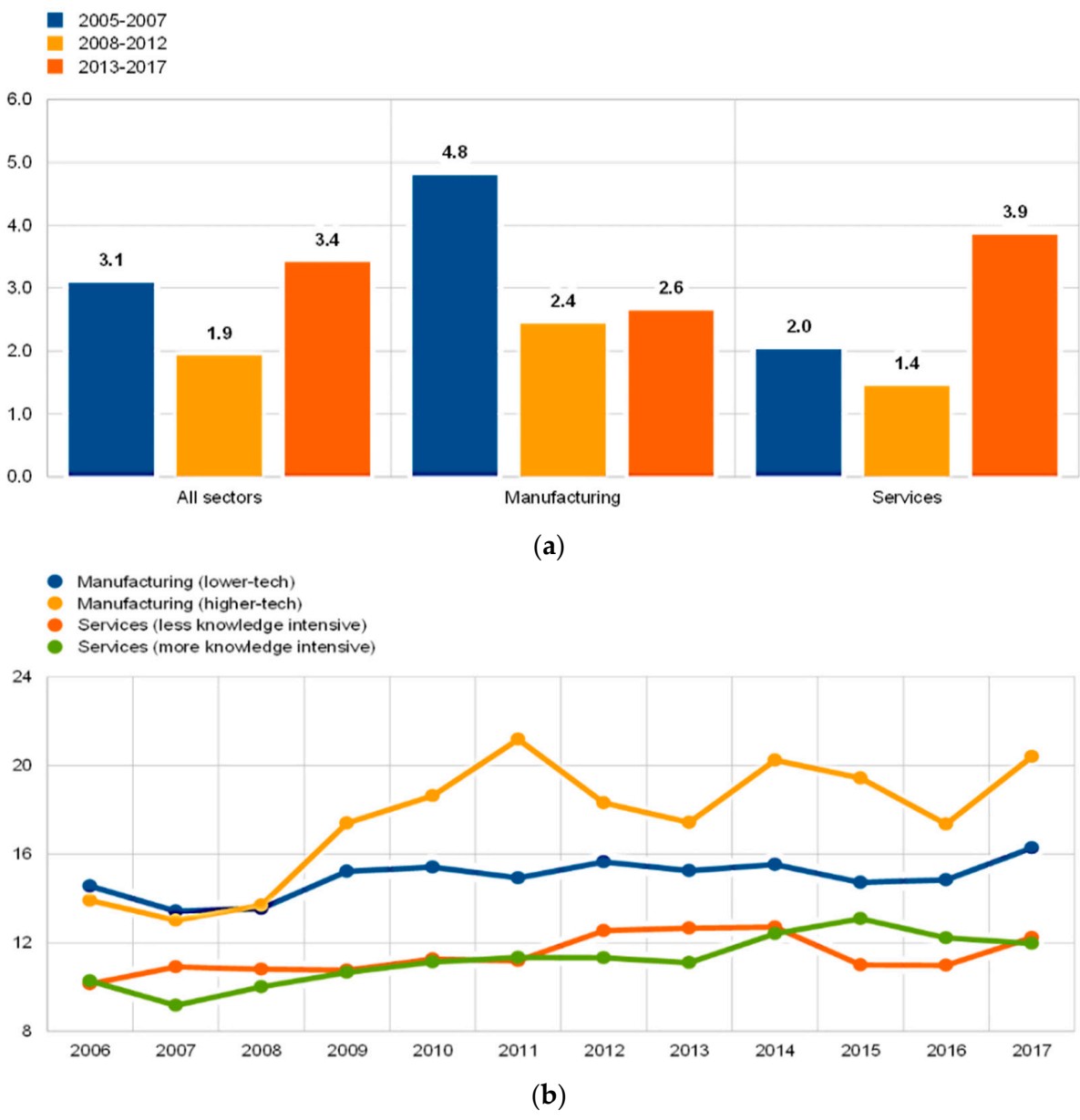

**Figure 3.** Productivity trends in the Euro area for 2006–2017. (**a**) TFP growth of frontier firms (the 5% of the most productive firms), averaged across countries. (**b**) Age of frontier firms by sector and technology intensity or knowledge intensity, averaged across countries for 2006–2017. Source: ECB Economic Bulletin, Issue 7/2021 [26].

In the first graph (Figure 3a), considering the top performers in the sector, the services sector showed the highest recovery after the financial crisis of 2008–2010, and the manufacturing did not succeed in regaining the position held before the same crisis. It is known that the sector of services was the most dynamic in the last decade, considering the contribution of the internet, digitalization, and ITC evolution.

In the second graph (Figure 3b), it can be observed that the manufacturing sector is more sensitive to innovation, while the services sector is not so sensitive to knowledge intensity.

At this point, it is possible to conclude that the services sector moved fast after the financial crisis of 2008–2010, with a large capacity for development after the crisis of 2020–2022. The current crisis is an opportunity to better deploy the value-added of knowledge in service sectors. At the same time, some fields which are more reluctant regarding knowledge assets (HORECA) have to include a deeper knowledge methodology in their business to keep pace with knowledge-intensive services fields, such as ITC.

On the manufacturing side, even though the recovery after the financial crisis of 2008–2010 was not so significant, in the subsequent years, there were dramatic changes in differentiating innovative companies from less open to innovative ones.

All in all, despite the adverse effects of the crises, there have been many opportunities to rethink the processes in terms of innovation, creativity, and entrepreneurship in such a way as to quickly adopt knowledge frontier standards and to move from physical resources to intangible ones.

## 4. Results and Discussion

Now, in times of deep and complex crises, this knowledge economy pyramid model of sustainable development is more reliable than the neo-liberal model because of the particularities of this complex environment: a pandemic for 3 years in a row, energy shocks, geopolitical conflict, and food security.

In this context, applying Schumpeter's theory of creative destruction, in which resources are reallocated to the more productive sectors, is not working conservatively. It is necessary to consider that many drivers of the current crisis do not come from market, economy, or financial disruption, but from health issues or geopolitical conflicts.

This is the reason why a collaborative environment is more reliable for rethinking the development model, where the government plays a critical role by investing in the most affected sectors, with implications for productivity, competitiveness, unemployment, inflation, and declining growth.

The COVID-19 crisis has had the most negative impact on global economies, similar to any other adverse events, such as after World War II, with bad effects on the demand side and the global supply chain. We are now in the midst of pandemic deployment, in the third year of the crisis, and we still do not know how the sanitary conditions are evolving. Despite this uncertain evolving situation, the characteristics of the disease are already known, the vaccination process is continuing, the teleworking regime is in place, mobility is adaptable, and we can manage many other factors to mitigate the effects. On the other side, the economic one, we see that the governments are deeply involved in keeping their economies open, many resources—especially financial ones—are available for recovery and resilience, and the socio-economic environment is ready for flexibility and adaptation to reduce the drawbacks and find development opportunities.

At the micro-level, the business environment has reacted with more adaptability by promoting innovation at the level of organizational strategy with a focus on creativity, using the digital advantages, adapting procedures, clarifying tasks, and elaborating a more flexible decision-making process. These all are the foundations of increasing productivity and competitiveness.

In a recent study [27], the researchers provided evidence that these measures could increase the productivity rate by 1% per year in the near future. In developed economies, this is two times higher than the rate of increase in productivity rate before the sanitary shock.

The effects of the coronavirus crisis were more visible in the labour factor of production, considering the accelerated implementation of digitalization in the workplace, public services, banking, education, and others. The lockdown more strongly affected those jobs where digitalization is more difficult to implement.

Regarding the effects of the crisis on the other production factor, capital, according to the observations for two pandemic years, the negative effects on the accumulation of physical capital are obvious.

Some of the sectors were more dynamic during this time, such as the ITC, health, research, education (especially higher education), and administration sectors. Considering this adaptability to the conditions of the pandemic, these sectors contribute more to increasing productivity than other sectors that were more affected by the crisis, such as agriculture, transportation, tourism, and HORECA.

At this point, a general observation could be that low-skilled jobs were more affected than jobs where the level of qualifications was higher. In other words, organizations with a lack of knowledge as a resource in the process were severely affected by the crisis.

As a consequence, as can be observed in the dynamism of digitalization during the crisis, another boosting factor is innovation. In order to increase productivity and competitiveness, it is necessary to empower innovation in production processes. This approach creates a foundation for long-term economic growth, on the one side, and increasing the resilience to the economic shocks on the other side.

According to this conclusion, it is necessary to understand how the specific conditions could affect performance in productivity: the differences between large corporations and SMEs, public and private enterprises, industry and agriculture, manufacturing and services, and so on. One of the most important drivers for mitigating these differences is the policy and regulatory environment. For instance, fostering investment in R&D, reallocating resources (especially human and financial), and highlighting the importance of education will have a positive impact on reconfiguring the competitive arena in the years to come.

Another approach is to consider reconfiguring the supply–demand balance to avoid breaches, bottlenecks, or other dysfunctionalities. At this moment, the semiconductor crisis, the energy crisis, and the lack of raw materials are some of the negative factors with a large impact on the level of macroeconomic stability, increasing inflation, unemployment, and poverty, and aggravating the imbalances at the social level.

Policymakers have to promote countercyclical measures to close the gaps created by the impact of COVID-19. Governments could play an active role in creating financing opportunities, managing fiscality, and making public investments in infrastructure, digitalization, the healthcare system, and education.

In the multilevel knowledge economy pyramid (MKEP) model, all these ecosystems work together in the same environment, and this will contribute to a better harmonization of research, education, business, and administration stakeholders, focusing on the reallocation of knowledge resources to increase productivity and competitiveness.

Without minimizing the role of labour and capital, in this research, the author tried to explain why knowledge has to be considered as an endogenous factor in the process of production. Moreover, it was demonstrated that during the pandemic crisis, processes that were more dependent on capital and labour were more affected than those based on knowledge. In other words, for sustainable development after a crisis, it is necessary to move the emphasis from hard to soft skills. This means technology, engineering, and digitalization are very important for increasing the level of performance, but from a knowledge perspective, these are just tools. Improving the knowledge process in parallel with the traditional ones will be the most efficient way to make progress in productivity and competitiveness in the near future.

In these circumstances, the focus has moved to the education system, technological advancement, the business environment, local communities, and administration, i.e., the stakeholders of the MKEP. All these have to be perceived as working together in a harmonized and complementary way in order to generate synergy. These actors have to jointly create what is called the knowledge environment. Starting from the bottom of the pyramid, knowledge workers, knowledge communities, and knowledge organizations should be created. This is the new approach proposed here, with the aim of improving performance.

It is necessary to make a critical shift from conservative thinking regarding factors such as workers, money, and machines, to visionary thinking that includes inputs such as knowledge, innovation, creativity, and entrepreneurship. In other words, the approach has moved from "what we have" to "what we can optimize with what we have". It is

necessary to reshape all the processes around knowledge as a primordial input in the production flow.

In other words, in a knowledge environment, the difference in productivity is not made by acquiring more workers, more money, and more machines, but by more knowledge on each side, and much more knowledge on all sides together.

Instead of having two operators who perform the same routine job, it is probably recommended to have one or two highly skilled workers who are educated and trained, able to innovate, create, and self-driven in collaboration with other individuals at the same level. The most important thing is not competition among individuals but cultivating complementarity.

There is a theory that urban areas are more productive than rural areas. Up to a certain point, this is true as long as the increased density of individuals creates a foundation for recruiting the best performers who have an impact on the outputs. More people gathered in the same place contribute to a knowledge-intensive process, considering the advantages of sharing knowledge, on-site learning, imitation, etc. [28]. The agglomeration of interactions is supposed to have a greater impact on productivity. However, letting more knowledge move in a Brownian manner is not as efficient as managing knowledge to create connections, better integration, and ecosystems. In terms of knowledge, it is important to manage quality, not quantity. Quantity is important at the input level in the knowledge process, such as education, learning, and skills. Quality in this process means innovation, creativity, and entrepreneurship.

A critical role in this process is played by the harmonization of the inputs (education, learning, and skills) with the level required by the jobs to avoid mismatching between the supply side and demand side of knowledge. A perfect calibrated knowledge process is one where both sides are optimally balanced, as it is shown in the Figure 4. A greater supply side in terms of knowledge, generates frustration, demotivation, and waste. A greater demand side means a lack of efficiency and effectiveness, as well as overwhelm and exhaustion.

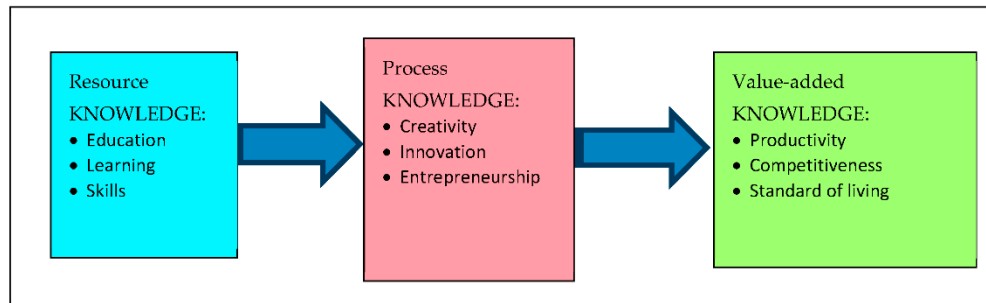

**Figure 4.** Perfect equilibrium between inputs and outputs in the process of knowledge. Source: Author's own work.

Beside innovation and physical assets for fostering productivity, an important role is played by intellectual capital, especially at the level of human resources, and the ability to capture knowledge from the production flows within an organization. Strong inputs for the consolidation of intellectual capital are coming from the education field. Creating a proactive platform for collaboration among universities, research institutes, and the business environment, in what is called the knowledge triangle, is the best response to the need to improve the level of human resources.

Another positive impact on increasing productivity comes from the capacity building area, and the ability of governments to create an institutional framework for the effectiveness of the productivity process. This involves several aspects: the institutional framework itself, the dedicated policy to be implemented, balanced interconnected legislation, managerial abilities, and network creation. Strong capacity building will determine good results in the productivity field.

Efforts to boost productivity are more efficient in a context where macroeconomic stability is consolidated, with a greater orientation towards investments into research, development, and innovation.

Usually, responsible governments are struggling to achieve macroeconomic stability with a solid commitment to sustainable growth, a high level of investment, a counter-cyclical economic policy, a low level of deficits and debts, high social protection, and many other actions to ensure the best functionality of a country economy [29]. This is not simple at all in the presence of competition in the international market, with diminishing access to resources, an increased population, and the more obvious effects of climate change.

Moreover, the full-time job of creating macroeconomic stability is transformed to an overwhelming task when governments are facing shocks. Throughout modern history, there have been several economic crises related to the finance and banking sector, oil and raw materials, and human resources and skills.

More disruptive crises are related to wars, pandemics, food security, and energy. Unfortunately, these days, it seems that all these crises are overlapping, with unexpected effects on the global economy, stability, peace, and geostrategic alliances.

## 5. Conclusions

During this research, the author found information, analysis, and explanations for the particularities of the crisis generated by COVID-19, geostrategic conflicts, energy shortages, and food security threats, which are associated with productivity and competitiveness. All these were simplified to the concept of the impact of adverse events on the economy. Considering the consequences of crisis on productivity and competitiveness, this study developed a new economic model, the multilevel knowledge economy pyramid, which is able to cope with the major adverse events acting at the global level in the last 3 years.

A knowledge environment is based on the four pillars of the knowledge economy and three structures—the knowledge triangle, the triple helix, and smart specialization—all unified by a central axis (education, the business environment, smart growth, and standards of living), with implications for value-added knowledge, productivity, competitiveness, and welfare. The conclusion is that competitiveness alone and individual strategies to win the battle against a complex crisis are not working any longer, and a more effective knowledge environment has to be created to overcome the crisis and to develop sustainable growth, where governments' involvement as partners in this game is more than necessary.

Currently, when pandemic crisis has not ended but other shocks are coming, including energy market disruptions, geopolitical instability, and food security threats, it is vital not only to have the commitment of national governments, but also to have strong alliances and unique voices from large regional organizations with great representativity in the global arena, such as the European Union's institutions, NATO, the US Administration, and all the other democratic regimes with similar values, objectives, and interests.

Considering these threats, the attitude towards productivity has to be changed in order to focus on enhancing the role of sustainable growth drivers, allocating more resources for research, creating a foundation for a green economy, implementing and extending digitalization at all levels, creating a flexible job environment, and creating a cooperative platform for socio-economic stakeholders.

These shocks are affecting productivity and have changed the framework of competitiveness. Without the commitment of governments, it will be nearly impossible to regain the pace of growth before the crisis in the short term. Any component of productivity is affected to some degree in times of crisis. For example:

- Innovation: restricted mobility, breaches on the global supply side, and a decline ion investments into R&D determine the disruptions in the adoption of new technologies, and this will affect TFP;
- Intellectual capital: the interruption of education programs, little connection with the real economy, the declining amplitude of learning programs, depopulation, and losing jobs determines the low accumulation of human capital.

- Capacity building: reallocating resources for critical sectors (health for the pandemic, the army for war, population subsidies for the energy crisis, and food security) weaken the institutional capacity in the field of productivity.
- Investments: sudden and large expenses for critical sectors will create shortages for investments in infrastructure, research, education, and the latest technology.
- Macroeconomic stability: negative growth rates will increase the structural deficits, debts, and inflation.

All these outputs of crisis require a mix of recovery and resilience strategies based on a huge injection of capital into the economy, keeping up with investment plans, ensuring an optimal macroeconomic balance, strengthening institutional capacity, direct protection of SMEs, and fostering educational programs.

Rethinking business processes in terms of using knowledge resources is an appropriate answer to address the crisis issues for several reasons, as follows:

- Creating a resilient and sustainable business environment: identifying, defining, mitigating, and managing risks; shortening the impact of shocks; and accelerating recovery;
- Better reallocation of resources (especially in terms of knowledge): understanding the nature of current value-added in processes that are more inclined towards the use of knowledge;
- Increasing productivity: process designs based on creativity, innovation, and entrepreneurship;
- Focusing on the transformation of intellectual capital: reducing routine work for individuals, and introducing robotics, automatization, digitalization, and AI.

The multilevel knowledge economy pyramid is a framework for future research into the evolving measurement system, Reference [30] such as the knowledge economy index, the competitiveness index, the innovation scoreboard, and many other composite indices.

The main advantage of the MKEP model for researchers is the ability to easily observe and understand all the connections and correlations, and their intensity, among the knowledge stakeholders involved in the efforts to increase productivity and competitiveness.

The MKEP model is a robust alternative to the neo-classical development model, where knowledge is managed in a logical framework capable of increasing the level of performance.

**Funding:** This research received no external funding.

**Institutional Review Board Statement:** Not applicable.

**Informed Consent Statement:** Not applicable.

**Data Availability Statement:** Not applicable.

**Conflicts of Interest:** The author declare no conflict of interest.

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
