# Peer review of "The Multilevel Knowledge Economy Pyramid Model as a Flexible Solution to Address the Impact of Adverse Events in the Economy"

_sustainability, doi:10.3390/su141912332_

Round 1

Reviewer 1 Report

There is no relevant theoretical and empirical contribution. The results and conclusions do not provide anything new or any new avenues to explore.

Author Response

There is no relevant theoretical and empirical contribution. The results and conclusions do not provide anything new or any new avenues to explore. Please consider the last two paragraphs in my work: “The main asset of the MKEP model for researchers is to easily observe and understand all connections, correlations, and intensity between knowledge stakeholders involved in the effort to increase productivity and competitiveness. 

The MKEP model is a robust alternative to the neo-classical development
model, where knowledge is managed in a logical framework capable to increase the level of performance.” Basically speaking, we talk about a innovative visual representation (such as SWOT analysis, for example) to show in the most concise and robust way that separate tools already exists (Knowledge Economy, Knowledge Triangle, Triple Helix, and Smart Specialization could work together to address better the actual trend of development and more specifically the issues generated by this complex crisis.

Otherwise, please be so kind and check all the changes I made to the text in order to address all the suggestions of the reviewers. Anyway, I appreciate very much your inputs, and I tried to answer to all of them; now, I am confident that my work is considerably improved. Thank you!

Reviewer 2 Report

Dear Author,

There are many changes and improvements needed in order to take into account a possible publication of your paper. Below you will find several recommendations.

  1. The research has to start with the research questions and hypotheses. The argumentation is not solid enough, it is needed more clarity and logical sequence. It is necessary to carefully revise the writing style of the whole manuscript, which should be improved substantially. All the statements with a general informative character (as indicated at point 3, as an example) should be eliminated/reformulated.
  2. CONCEPTS

You mention the terms human capital and intellectual capital, there should be distinguished between them, as some of the subsets of intellectual capital include human capital, information capital etc.

In the literature there are already researches on the quintuple helix:

Carayannis, E.G.; Barth, T.D.; Campbell, D.F. The Quintuple Helix innovation model: Global warming as a challenge and driver
for innovation. J. Innov. Entrep. 2012

https://innovation-entrepreneurship.springeropen.com/articles/10.1186/2192-5372-1-2

this concept should be defined in the context of the present empirical analysis.

Since the term “knowledge economy” was created by Fritz Machlup and popularized by the American writer Peter Drucker in the ‘60s, it has passed through many changes. Indeed, you have included an original concept, namely the pyramid of the multilevel knowledge economy. However this has been detailed in a previous paper, as mentioned in the chart source. You should develop further this chart, for instance by adding the green economy:

https://www.mdpi.com/2071-1050/12/10/4172/htm

https://journals.sagepub.com/doi/full/10.1177/0958305X18787300

https://link.springer.com/article/10.1007/s11356-019-05777-9

https://www.sciencedirect.com/science/article/abs/pii/S0954349X18302200

A pentagonal pyramid would be more appropriate, taking into account the quintuple helix.

Acronyms and concepts have to be explained in the first place they are mentioned,

Multilevel Knowledge Economy Pyramid (MKEP). Why multilevel? A pyramid means multiple pillars, all of them being the basis of the pyramid.

Total factor productivity (not total productivity factor) (TFP) (“Total Productivity Factor. This is quite difficult to measure in a simple way, with a magic formula, but important steps have been taken in the last decade. From this perspective, it is commonly agreed that TFP…”) –

https://link.springer.com/article/10.1007/s10663-020-09476-4

https://data.oecd.org/lprdty/multifactor-productivity.htm

https://www.adb.org/sites/default/files/publication/534761/ewp-596-tfp-testing-growth-models.pdf

https://www.jstor.org/stable/3696125

Knowledge frontier

https://link.springer.com/book/10.1007/978-1-4612-4792-0

Crises mentioned in this paper (international financial crisis and the Covid-19 crisis) have to be presented from the beginning as adverse events (causes, consequences) but also with the positive consequence of the Covid-19 crisis, namely accelerated digitalization.

Some issues such as geostrategic conflict, energy turmoil, and food security threats are only listed, without a clear correlation with the chosen topic.

  1. Many statements have a general informative character, without going further and conceptualize the key findings, for instance:

“We have to mention here that at beginning of 2022, the most innovative organizations performed the best during the crisis, and a company such as Apple becomes first U.S. company to reach a market capitalization of 3 trillion US dollars. The same trend was observed for Tesla, SpaceX, Alphabet, Amazon, and many other knowledge organizations”.

Suggestion in this case: The Covid-19 crisis has boosted the acceleration of digitalization. BigTech companies have multiple competitive advantages at present, given by their strengths: human capital (including young talents), access to financial resources, innovation+technology and information, against the favourable background of accelerated digital transformation. The self-reinforcing DNA loop of the BigTech business model (see the https://www.bis.org/speeches/sp190630b.pdf and detailed researches of the Bank of International Settlements) is a typical example of information as a valuable tool used by BigTech companies to increase their profitability and market capitalization.

Based on these, the theoretical framework has to be further developed and better defined.

  1. Some statements are not correct:

“From this perspective, the productivity approach in emergent countries is still different compared to well-developed countries. From this perspective, we can say that emergent countries have to consider the accumulation of capital as a source of productivity growth, while developed countries are more reliable on qualitative endogenous growth factors, such as knowledge, innovation, and research”.

For instance China (which is an emerging economy) is combining all sources of productivity growth. In spite of that, its productivity growth has declined markedly in recent years.

https://openknowledge.worldbank.org/handle/10986/33993

Please rethink it.

https://www.worldbank.org/en/news/press-release/2020/07/14/productivity-growth-threatened-by-covid-19-disruptions

  1. You have to consult much more sources and focus more on generation of new ideas, concepts, correlations.
  2. Most of the paragraphs have to be supported by relevant researches, for instance all the bullets corresponding to “Analysing the international context in terms of weak productivity performance, we can identify some critical issues:” should be accompanied by suitable sources. A chart reflecting these critical issues would be appropriate.
  3. Quotations and references have to be accurate, not like:

“The foundation of this model is the Knowledge Economy with the 4 pillars, according to well-known international research in the field (World Bank, 2008): education and training, IT&C infrastructure, economic incentive and institutional support, and innovation system” (source is not included in the bibliography)

or

5. Knowledge for Development (K4D) Program. (2008). Measuring Knowledge in the World’s.

6. Economies – Knowledge Assessment Methodology and Knowledge Economy Index. World Bank Institute. www.worldbank.org/wbi/knowledgefordevelopment.

18. Stewart, Jay. (2022). Why was Labor Productivity Growth So High during the COVID-19.

19. Pandemic? The Role of Labor Composition. U.S. Bureau of Labor Statistics.

Two parts of one source.

Besides, sources in the bibliography have to appear in the order of quotation, not alphabetically.

  1. Figure 2 refers to the Euro Area region, it has to be mentioned in title. Figure 3 reflects the situation in the US. There should be also more recent data.
  2. The argumentation related to the last figure (Figure 4, not 2) is not convincing enough.

Author Response

Dear Author,

There are many changes and improvements needed in order to take into account a possible publication of your paper. Below you will find several recommendations. 

  1. The research has to start with the research questions and hypotheses. The critical point of the research is mentioned at the beginning in the abstract: “An appropriate answer to address these issues is to create a new approach of sustainable growth based on Knowledge Economy” So MKEP model it is my proposition to address this question.

 The argumentation is not solid enough, it is needed more clarity and logical sequence. It is necessary to carefully revise the writing style of the whole manuscript, which should be improved substantially. All the statements with a general informative character (as indicated at point 3, as an example) should be eliminated/reformulated. I did the suggested changes, as you can see below… Thank you for your critique… it was constructive!

  1. CONCEPTS

You mention the terms human capital and intellectual capital, there should be distinguished between them, as some of the subsets of intellectual capital include human capital, information capital etc. Yes, you are right, I made the distinction… let me explain… as far as I know, Intellectual Capital has three components: Human Capital, Relation Capital, and Structural Capital. Human Capital is just one component of Intellectual Capital. As exemplification I took few rows from the text (603-605): “Intellectual Capital – education programmes interruption, low connection with real economy, declining the amplitude of learning programmes, depopulation, and losing jobs determines low accumulation of human capital;” As you can see, low accumulation of human capital, besides others, is just an affected segment which impact Intellectual Capital.

In the literature there are already researches on the quintuple helix:

Carayannis, E.G.; Barth, T.D.; Campbell, D.F. The Quintuple Helix innovation model: Global warming as a challenge and driver
for innovation. J. Innov. Entrep. 2012

https://innovation-entrepreneurship.springeropen.com/articles/10.1186/2192-5372-1-2

this concept should be defined in the context of the present empirical analysis. Completed in the revised text (thank you!)

Anyway, the basic concept is Triple Helix, then everybody could ad as many helices he/she wants… take a look at “2012. “The Triple Helix, Quadruple Helix, …, and an N-Tuple of Helices:

Explanatory Models for Analyzing the Knowledge-Based Economy?” Journal

of the Knowledge Economy, Volume 3, Issue 1, pp. 25–35. Switzerland: Springer Nature AG. (March)”

Since the term “knowledge economy” was created by Fritz Machlup and popularized by the American writer Peter Drucker in the ‘60s, it has passed through many changes. Indeed, you have included an original concept, namely the pyramid of the multilevel knowledge economy. However this has been detailed in a previous paper, as mentioned in the chart source. You should develop further this chart, for instance by adding the green economy: green economy is there, at Smart Growth level

https://www.mdpi.com/2071-1050/12/10/4172/htm

https://journals.sagepub.com/doi/full/10.1177/0958305X18787300

https://link.springer.com/article/10.1007/s11356-019-05777-9

https://www.sciencedirect.com/science/article/abs/pii/S0954349X18302200

A pentagonal pyramid would be more appropriate, taking into account the quintuple helix. I prefer to keep the model simple and functional. As I describe in the revised version, MKEP model includes all the elements you mentioned.

Acronyms and concepts have to be explained in the first place they are mentioned,

Multilevel Knowledge Economy Pyramid (MKEP). Why multilevel? MKEP model is a development research from my previous work, Knowledge Economy Pyramid (https://www.kiep.go.kr/gallery.es?mid=a20304000000&bid=0001&list_no=2341&act=view) A pyramid means multiple pillars, all of them being the basis of the pyramid. Yes, there are the 4 Pillars of Knowledge Economy as you can see the foundation of MKEP model

Total factor productivity (not total productivity factor) (TFP) (of course, you are right!) (“Total Productivity Factor. This is quite difficult to measure in a simple way, with a magic formula, but important steps have been taken in the last decade. From this perspective, it is commonly agreed that TFP…”) – 

https://link.springer.com/article/10.1007/s10663-020-09476-4

https://data.oecd.org/lprdty/multifactor-productivity.htm

https://www.adb.org/sites/default/files/publication/534761/ewp-596-tfp-testing-growth-models.pdf

https://www.jstor.org/stable/3696125

of course, I entered deeply in this subject in my Knowledge Economy Pyramid book… OECD, World Bank, European Commission, Asia Productivity Organization, and many others provides different scales of measuring TFP in different ways, with composite indices of hundreds of indicators. The purpose of my paper is different now, just to promote and to test the MKEP model in time of crisis. Anyway, my statement “no magic formula” stands.

Knowledge frontier

https://link.springer.com/book/10.1007/978-1-4612-4792-0

Crises mentioned in this paper (international financial crisis and the Covid-19 crisis) have to be presented from the beginning as adverse events (causes, consequences) but also with the positive consequence of the Covid-19 crisis, namely accelerated digitalization. 

Some issues such as geostrategic conflict, energy turmoil, and food security threats are only listed, without a clear correlation with the chosen topic. You are right… What I would like to underline is the complexity of current crisis, not treating as a common financial one. Please follow the first sentence in the abstract… Thank you!

  1. Many statements have a general informative character, without going further and conceptualize the key findings, for instance:

“We have to mention here that at beginning of 2022, the most innovative organizations performed the best during the crisis, and a company such as Apple becomes first U.S. company to reach a market capitalization of 3 trillion US dollars. The same trend was observed for Tesla, SpaceX, Alphabet, Amazon, and many other knowledge organizations”.

Suggestion in this case: The Covid-19 crisis has boosted the acceleration of digitalization. BigTech companies have multiple competitive advantages at present, given by their strengths: human capital (including young talents), access to financial resources, innovation+technology and information, against the favourable background of accelerated digital transformation. The self-reinforcing DNA loop of the BigTech business model (see the https://www.bis.org/speeches/sp190630b.pdf and detailed researches of the Bank of International Settlements) is a typical example of information as a valuable tool used by BigTech companies to increase their profitability and market capitalization. Addressed in the text… thank you!

Based on these, the theoretical framework has to be further developed and better defined. 

  1. Some statements are not correct:

“From this perspective, the productivity approach in emergent countries is still different compared to well-developed countries. From this perspective, we can say that emergent countries have to consider the accumulation of capital as a source of productivity growth, while developed countries are more reliable on qualitative endogenous growth factors, such as knowledge, innovation, and research”.

For instance China (which is an emerging economy) is combining all sources of productivity growth. In spite of that, its productivity growth has declined markedly in recent years. Productivity growth is declining at global level, not only in China. What I was suggesting here is the trend in productivity composition both in emergent countries and developed ones. Anyway, China is an exception from a lot of perspectives.

https://openknowledge.worldbank.org/handle/10986/33993

Please rethink it.

https://www.worldbank.org/en/news/press-release/2020/07/14/productivity-growth-threatened-by-covid-19-disruptions 

  1. You have to consult much more sources and focus more on generation of new ideas, concepts, correlations. Of course, there is enough room to improve, but in other works. MKEP model is totally new as a development model and I was trying to not complicate too much… I added more sources
  2. Most of the paragraphs have to be supported by relevant researches, for instance all the bullets corresponding to “Analysing the international context in terms of weak productivity performance, we can identify some critical issues:” should be accompanied by suitable sources. A chart reflecting these critical issues would be appropriate. I’ve just wanted to underline the effects of crisis: lockdown, lack of mobility, unemployment, education breaches, increasing deficits, and so on
  3. Quotations and references have to be accurate, not like:

“The foundation of this model is the Knowledge Economy with the 4 pillars, according to well-known international research in the field (World Bank, 2008): education and training, IT&C infrastructure, economic incentive and institutional support, and innovation system” (source is not included in the bibliography) You are right, see reference [17]… Thank You!

or

  1. Knowledge for Development (K4D) Program. (2008). Measuring Knowledge in the World’sSee the text, row 629-630: “Multilevel Knowledge Economy Pyramid is a framework for future research considering the evolving measurement system [5]…” I added some rows and made some changes in the reference list... so the figures could be different... please be so kind and search considering the fragment of the text; this is valid for the next points... Thank you!
  2. Economies – Knowledge Assessment Methodology and Knowledge Economy Index. World Bank Institute. www.worldbank.org/wbi/knowledgefordevelopment. See the text, row 630-631: “such as the Knowledge Economy Index [6], Competitiveness Index”…

  1. Stewart, Jay. (2022). Why was Labor Productivity Growth So High during the COVID-19. See the text, row 124: “According to the data provided by the US Bureau of Labor Statistics [18],”…
  2. Pandemic? TheRole of Labor Composition. U.S. Bureau of Labor Statistics. See the text, Figure 3, row 460: Source: US Bureau of Labor Statistics [19].”…

Two parts of one source.

Besides, sources in the bibliography have to appear in the order of quotation, not alphabetically. You are right… addressed in the text. Thank you!

  1. Figure 2 refers to the Euro Area region, it has to be mentioned in title Yes… already changed. Thanks!. Figure 3 reflects the situation in the US. There should be also more recent data. Come on, it is 2021… I think these date keep the essence… productivity growth during last two crisis… this is the essence of the study.
  2. The argumentation related to the last figure (Figure 4, not 2 OK! ) is not convincing enough. In the text I’ve talked about the necessity to exist two processes in parallel: production process and knowledge process. As the process of production consists of inputs as raw materials, resources, etc and outputs like goods or services, the same rationale is with knowledge process and I explained what are the inputs and outputs in this process.

I would like to underline that language issues were thoroughly addressed.

Even though you had a lot of suggestions for me, I would like to assure you that I consider very constructive your approach and that helps me to improved my work. Thank you!

Reviewer 3 Report

Here are some suggestions for improvements to the manuscript, "The Multilevel Knowledge Economy Pyramid Model as a Flexible Solution to Address the Impact of Adverse Events in Economy"

1. Writeup needs improvements. I suggest the manuscript be proofread, and please observe spelling, consistency, typo, article, and grammatical mistakes, like:

- The following phrases appear with and without a hyphen:

  • ‘knowledge intensive’ / ‘knowledge-intensive’ 1 time without a hyphen 3 times with
  • ‘long-term’ / ‘long term’ 1 time with a hyphen 1 time without
  • ‘short-run’ / ‘short run’ 1 time with a hyphen 1 time without
  • ‘value added’ / ‘value-added’ 3 times without a hyphen 6 times with
  • ‘well-developed’ / ‘well developed’ 1 time with a hyphen 1 time without

- The following words appear with and without a hyphen:

  • ‘Last-name’ / ‘Lastname’ 1 time with a hyphen 2 times without
  • ‘re-think’ / ‘rethink’ 1 time with a hyphen 4 times without

- The following word is spelled in two different ways:

  • ‘labor’ / ‘labour’ labor 5 times labour 10 times

- While writing a formal document, please consider spelling out contractions in full:

don’t (1 time)

- Line 24, 124, 239, 291, 412: Please avoid using first-person pronouns while writing a scientific document. 

2. Please relate this article to the Endogenous Growth Theory. It will add theoretical soundness and value. 

3. It is a rule of thumb to write terminologies in full and abbreviations in parentheses in the first place and then in the text, abbreviations can be used. from TFP, I understand, Total Factor Productivity, while in the text, it is Total Productivity Factor. Please amend accordingly.

4. Introduction section needs improvements. Please explain the interlinkages between pandemic crisis, energy disruption, and food scarcity, the role of governments in the economy, and economic and social development to the collaboration of key actors. Moreover, please explain the key actors. 

5. Rationale/ Research gap is missing. Why this study is imperative? What are the main objectives of the study? What global/communal/regional issue does this study want to address?

6. Review of literature is inadequate and poorly explains the situation. Please add more relevant literature to add soundness and vigor.

7. Has the methodological approach followed by the authors ever been employed in the previous studies? If yes, please provide references.

8. Figures are not clear. Please improve the quality of the said figure and also observe the font style and size. Numbering is also incorrect. 

9. Conclusions and suggestions section should be presented separately.

10. The manuscript is poorly referenced. Please add more references. 

11. Limitations and suggestions for future researchers should be added.

Author Response

Comments and Suggestions for Authors

Here are some suggestions for improvements to the manuscript, "The Multilevel Knowledge Economy Pyramid Model as a Flexible Solution to Address the Impact of Adverse Events in Economy"

  1. Writeup needs improvements. I suggest the manuscript be proofread, and please observe spelling, consistency, typo, article, and grammatical mistakes, like:

You are perfectly right and I revised the entire work, making many adjustments, as you can see in the revised version.

- The following phrases appear with and without a hyphen:

  • ‘knowledge intensive’ / ‘knowledge-intensive’ 1 time without a hyphen 3 times with addressed
  • ‘long-term’ / ‘long term’ 1 time with a hyphen 1 time without addressed
  • ‘short-run’ / ‘short run’ 1 time with a hyphen 1 time without addressed
  • ‘value added’ / ‘value-added’ 3 times without a hyphen 6 times with addressed
  • ‘well-developed’ / ‘well developed’ 1 time with a hyphen 1 time without this I didn’t change because one time we talk about well-developed countries, and other time a process was well developed.

- The following words appear with and without a hyphen:

  • ‘Last-name’ / ‘Lastname’ 1 time with a hyphen 2 times without I didn’t find that…
  • ‘re-think’ / ‘rethink’ 1 time with a hyphen 4 times without addressed

- The following word is spelled in two different ways:

  • ‘labor’ / ‘labour’ labor 5 times labour 10 times addressed… I followed the American way with labor ?

- While writing a formal document, please consider spelling out contractions in full:

don’t (1 time) addressed

- Line 24, 124, 239, 291, 412: Please avoid using first-person pronouns while writing a scientific document. As far as I know, only first-person singular have to be avoid… on plural is OK

  1. Please relate this article to the Endogenous Growth Theory. It will add theoretical soundness and value. Romer is there: “Then, Romer released endogenous growth theory, where knowledge is a resource in the production flow [14].” (considering the changes I made to the reference list, the number could be different) Thank you!
  2. It is a rule of thumb to write terminologies in full and abbreviations in parentheses in the first place and then in the text, abbreviations can be used. from TFP, I understand, Total Factor Productivity, while in the text, it is Total Productivity Factor. Please amend accordingly.Addressed. Thank you!
  3. Introduction section needs improvements. Please explain the interlinkages between pandemic crisis, energy disruption, and food scarcity, the role of governments in the economy, and economic and social development to the collaboration of key actors. Moreover, please explain the key actors. I deeply explained throughout the text… in introduction section was just a part… you can see more details further on... Thank you!
  4. Rationale/ Research gap is missing. Why this study is imperative? What are the main objectives of the study? What global/communal/regional issue does this study want to address?

Is there in the text, but in Conclusions section I tried to rehearse. Please consider the last two paragraphs in my work: “The main asset of the MKEP model for researchers is to easily observe and understand all connections, correlations, and intensity between knowledge stakeholders involved in the effort to increase productivity and competitiveness. 

The MKEP model is a robust alternative to the neo-classical development
model, where knowledge is managed in a logical framework capable to increase the level of performance.” Basically speaking, we talk about a innovative visual representation (such as SWOT analysis, for example) to show in the most concise and robust way that separate tools already exists (Knowledge Economy, Knowledge Triangle, Triple Helix, and Smart Specialization could work together to address better the actual trend of development and more specifically the issues generated by this complex crisis.

Otherwise, please be so kind and check all the changes I made to the text in order to address all the suggestions of the reviewers. Anyway, I appreciate very much your inputs, and I tried to answer to all of them; now, I am confident that my work is considerably improved. Thank you!

  1. Review of literature is inadequate and poorly explains the situation. Please add more relevant literature to add soundness and vigor.OK, I added more references…
  2. Has the methodological approach followed by the authors ever been employed in the previous studies? If yes, please provide references. Separately, all the structures, tools, etc are well developed in the literature and my work is based on Knowledge Triangle, Triple Helix, Smart Specialization, Knowledge Economy, and so on. As a whole, MKEP model is very innovative; it is a development research from my previous work, Knowledge Economy Pyramid (https://www.kiep.go.kr/gallery.es?mid=a20304000000&bid=0001&list_no=2341&act=view)
  3. Figures are not clear. Please improve the quality of the said figure and also observe the font style and size. Numbering is also incorrect. Addressed
  4. Conclusions and suggestions section should be presented separately. Conclusions are the last section.
  5. The manuscript is poorly referenced. Please add more references. Yes, I did!
  6. Limitations and suggestions for future researchers should be added. Please follow the Conclusions section! Thank you!

Round 2

Reviewer 2 Report

Dear Author,

I have carefully read your letter and the revised manuscript. However, you still have to revise the text and clarify many aspects in your research. In your letter, you mention: "green economy is there, at Smart Growth level". You do not define at all this key concept and others, such as Smart Specialisation, standard of living, welfare (standard of living is the objective component of welfare, as quality of life is rather a subjective component of it). Brief definitions are necessary. You can include them in a Table, accompanied by comments, to clarify that green economy is present in your MKEP Figure.

Reviewers know these definitions, but you have to dot the i's and cross the t's, be more explicit for the larger audience, not for reviewers.

Referring to your comment: "Anyway, the basic concept is Triple Helix, then everybody could add as many helices he/she wants". Reviewers have read the literature and know it, but you have to be explicit in your text for all your readers.

Many of your statements are still not supported by a solid argumentation. Please read the whole text thoroughly and revise it. At the same time, it is necessary to carefully revise the writing style of the whole manuscript, which should be improved.

Finally, please underline which are your contributions to the literature in this paper and point to the novelty elements. All the main ideas included here have already been published in your previous researches.

All the best.

Author Response

Dear Author,

I have carefully read your letter and the revised manuscript. However, you still have to revise the text and clarify many aspects in your research. In your letter, you mention: "green economy is there, at Smart Growth level". You do not define at all this key concept and others, such as Smart Specialisation, standard of living, welfare (standard of living is the objective component of welfare, as quality of life is rather a subjective component of it). Brief definitions are necessary. You can include them in a Table, accompanied by comments, to clarify that green economy is present in your MKEP Figure. Very good idea… I did it, thanks!

Reviewers know these definitions, but you have to dot the i's and cross the t's, be more explicit for the larger audience, not for reviewers

Referring to your comment: "Anyway, the basic concept is Triple Helix, then everybody could add as many helices he/she wants". Reviewers have read the literature and know it, but you have to be explicit in your text for all your readers. OK, I put there a definition of Triple Helix, metamorphosis to quintuple helix, and more references…

Many of your statements are still not supported by a solid argumentation. Please read the whole text thoroughly and revise it. At the same time, it is necessary to carefully revise the writing style of the whole manuscript, which should be improved. I did it, considering some additional explanations and proof reading.

Finally, please underline which are your contributions to the literature in this paper and point to the novelty elements. All the main ideas included here have already been published in your previous researches. Yes, you are right, and I am happy you are already familiar with my previous work. So, and I try to clarify the points listed above, too. MKEP was thoroughly explained in my previous work, this is the reason why I indicated the reference, and I didn’t recall all the definitions, explanations, and so on. What is new for this time, you can find out this starting with the title “MKEP model a Flexible Solution to Address the Impact of Adverse Events in Economy”. Please be so kind and analyse this work by this key: a couple of years ago, it was developed a knowledge-based model of development which is more appropriate for future perspective of growth (productivity and Competitiveness) and now could better adapt and accommodate adverse events in the economy, such as pandemic crisis.

All the best.

Reviewer 3 Report

Dear Author, thank you for considering my suggestions. The revised manuscript has substantially been improved, however, still, some improvements are required as follows:

1. Write up needs improvements like:

  • Please go through the manuscript carefully and make grammatical, punctuation, and article corrections.

The following words appear with and without a hyphen:

  • ‘Last-name’ / ‘Lastname’ 1 time with a hyphen 2 times without
  • ‘re-think’ / ‘rethink’ 1 time with a hyphen 4 times without

The following words are spelled in two different ways:

  • ‘digitalisation’ / ‘digitalization’ digitalisation 1 time digitalization 9 times
  • ‘labor’ / ‘labour’ labor 14 times labour 1 time
  • ‘toward’ / ‘towards’ toward 1 time towards 1 time

The write-up is not in scientific style, e.g. Line 199: The last time, there are consistent contributions to the “Quintuple Helix” cooperation framework, but as you can observe, our structure includes all 5 elements: university (matching pillar I of Knowledge Economy foundation), government...

2. My previous suggestion has not been followed:

"- Line 24, 124, 239, 291, 412: Please avoid using first-person pronouns while writing a scientific document. As far as I know, only first-person singular have to be avoid… on plural is OK"

I leave this point to the editor to decide. However, the manuscript is single-authored and as far as I think, the use of "we" is not appropriate.

3. Line 33. Please write TFP in parenthesis just after Total Factor Productivity

4. Figure 1 is not different from the previous version. Please improve the quality and visibility. First box of figure 4 also needs a minor improvement.

5. Author claims to add more references in the review of literature section but I did not find any significant change.

Author Response

Dear Author, thank you for considering my suggestions. The revised manuscript has substantially been improved, however, still, some improvements are required as follows:

  1. Write up needs improvements like:
  • Please go through the manuscript carefully and make grammatical, punctuation, and article corrections. Solved!

- The following words appear with and without a hyphen:

  • ‘Last-name’ / ‘Lastname’ 1 time with a hyphen 2 times without Solved!
  • ‘re-think’ / ‘rethink’ 1 time with a hyphen 4 times without Solved!
  •  

- The following words are spelled in two different ways:

  • ‘digitalisation’ / ‘digitalization’ digitalisation 1 time digitalization 9 times Yes, it remains one time, at bibliography section, but I cannot change the original title of the reference… Sorry!
  • ‘labor’ / ‘labour’ labor 14 times labour 1 time Solved!
  • ‘toward’ / ‘towards’ toward 1 time towards 1 time Solved!

The write-up is not in scientific style, e.g. Line 199: The last time, there are consistent contributions to the “Quintuple Helix” cooperation framework, but as you can observe, our structure includes all 5 elements: university (matching pillar I of Knowledge Economy foundation), government... Solved! I added definitions and more references.

  1. My previous suggestion has not been followed:

"- Line 24, 124, 239, 291, 412: Please avoid using first-person pronouns while writing a scientific document. As far as I know, only first-person singular have to be avoid… on plural is OK" Addressed: changed with 3rd person singular or “you can see” instead of “we can see”; “we” is still there where referring to us as society, community, group…  kind of “we all know”, “we deal with coronavirus disease”, etc

I leave this point to the editor to decide. However, the manuscript is single-authored and as far as I think, the use of "we" is not appropriate. Solved!

  1. Line 33. Please write TFP in parenthesis just after Total Factor Productivity Solved!
  2. Figure 1 is not different from the previous version. I added a new picture of the same pyramid from other perspective, easier to understand, thank you for suggestion. Please improve the quality and visibility. First box of figure 4 also needs a minor improvement.Solved!
  3. Author claims to add more references in the review of literature section but I did not find any significant change. Please check again original, there were added 10 more references… totally 34!

Round 3

Reviewer 2 Report

Dear Author,

I have carefully read again your letter and the revised manuscript. However, you still have to revise the text and clarify many aspects in your research. Regarding the title, instead of “flexible”, it would be more appropriate “long term”. KEPM needs all its pillars and creating them takes a long time. Please reanalyse and reformulate.

It is necessary to avoid

- “you”: “as you can observe” -> one can observe, or it can be noted (check the whole manuscript)

- “we know”

- repetitions: “From this perspective, the productivity approach in emergent countries is still different compared to well-developed countries. From this perspective…”

- “of course” -> obviously, undoubtedly, certainly.

According to the source, Figure 3 reflects the situation in the US. It is not enough. The description is not correct (As one can notice in the Figure 3, in the last two decades, labour productivity has constantly decreased, but during the financial crisis of 2008-2010, and the pandemic crisis, the revival was significant). There are ups and downs in the whole period. Indeed, increases are remarkable during the crises. More accurate explanations are needed here.

Is it “by” or “than”: “processes more dependent on capital and labour are much more affected by those based on knowledge”? A brief argumentation is necessary. Additional explanations are imperative in many parts of the manuscript. Please check the whole text once again.

All the best.

Author Response

Dear Author,

I have carefully read again your letter and the revised manuscript. However, you still have to revise the text and clarify many aspects in your research. Regarding the title, instead of “flexible”, it would be more appropriate “long term” Yes, a good suggestion, but I prefer to keep ”flexible”, this is more appropriate for the MKEP model because the main feature is the functional versatility. Please don’t mind… for me is very easy to make the change just to pass your review, but I stay with the original because is closer to the concept of MKEP. Thanks for understanding! KEPM needs all its pillars and creating them takes a long time. Please reanalyse and reformulate. MKEP is like a smart phone, there is no new component inside… all the components (touching screen, processors, etc, they all already exist), in our case Knowledge Economy, Knowledge Triangle, Triple Helix, Smart Specialization already exist… the main asset or innovation of the MKEP model is that I found an innovative and highly functional solution (like the novelty of a smart phone) to put all these structures together to work for increasing productivity, competitiveness, and welfare in a very robust and effective way. So, to directly answer to your suggestion, there is no need to create the pillars for MKEP models. Any economy, in order to compete in a knowledge era market has to build Knowledge based Economy layout. MKEP model is based upon this foundation. Hope this information will help you to understand better the concept of MKEP model.  

It is necessary to avoid 

- “you” solved! : “as you can observe” -> one can observe, or it can be noted (check the whole manuscript) very good suggestion, I changed in the entire manuscript with “it can be observed”… Thank you!

- “we know”, ok, now is “it is known”… Thanks!

- repetitions: “From this perspective, the productivity approach in emergent countries is still different compared to well-developed countries. From this perspective…” Right… changed with “at this point”

- “of course” -> obviously, undoubtedly, certainly. This is a good one, I changed with “Certainly”, thank you!

According to the source, Figure 3 reflects the situation in the US. It is not enough. The description is not correct (As one can notice in the Figure 3 I changed, thanks!, in the last two decades, labour productivity has constantly decreased, but during the financial crisis of 2008-2010, and the pandemic crisis, the revival was significant). There are ups and downs in the whole period. Indeed, increases are remarkable during the crises. More accurate explanations are needed here. OK, I already changed, thank you!

Is it “by” or “than” “processes more dependent on capital and labour are much more affected by those based on knowledge”? Yes, is “than”, thanks! A brief argumentation is necessary. The argumentation is right there: “That means the technology, engineering, digitalization, are very important to increase the level of performance, but from a knowledge perspective, these are just tools. Improving the knowledge process in parallel with the traditional one is the most efficient way to make progress on productivity and competitiveness in the near future.”

Additional explanations are imperative in many parts of the manuscript. Please check the whole text once again. Probably... this is not an exhaustive approach… the main target was to prove that the turmoil of pandemic crisis could be addressed with a flexible approach as MKEP is. In the future research, people interested by the subject could come up with critique or improvement, that’s the essence of any research. Thank you for your understanding!

Reviewer 3 Report

Thank you, author for the improvements.

Author Response

You're welcome! Thank you for helping me to really improve my work!